# Cumulative Effects of Genetic Variants Detected in a Child with Early-Onset Non-Syndromic Obesity Due to SIM-1 Gene Mutation

**DOI:** 10.3390/genes16050588

**Published:** 2025-05-17

**Authors:** Giovanni Luppino, Malgorzata Wasniewska, Mara Giordano, Giorgia Pepe, Letteria Anna Morabito, Debora Porri, Tommaso Aversa, Domenico Corica

**Affiliations:** 1Department of Human Pathology of Adulthood and Childhood, University of Messina, Via Consolare Valeria 1, 98125 Messina, Italy; giovilup97@gmail.com (G.L.); giorgia.pepe@unime.it (G.P.); debora.porri@gmail.com (D.P.); tommaso.aversa@unime.it (T.A.); domenico.corica@unime.it (D.C.); 2Pediatric Unit, AOU Policlinico G. Martino, Via Consolare Valeria 1, 98125 Messina, Italy; letteria.morabito@gmail.com; 3Laboratory of Genetics, Struttura Complessa a Direzione Universitaria (SCDU) Biochimica Clinica, Ospedale Maggiore della Carità, 28100 Novara, Italy; mara.giordano@med.uniupo.it; 4Department of Health Sciences, University of Piemonte Orientale, 28100 Novara, Italy

**Keywords:** early onset obesity, hyperphagia, *SIM1* gene, *SEMA3* gene, *PLXNA* gene, *CREBBP* gene, neurobehavior disorders

## Abstract

Background: Single-minded homolog 1 (*SIM1*) gene mutations with autosomal dominant inheritance have been related to hyperphagia and early-onset severe obesity. *SIM1* is implicated in the development of hypothalamic nuclei, which play a crucial role in energy homeostasis. The development of melanocortin neural circuits in the hypothalamus is promoted by other factors such as Semaphorine 3 (*SEMA3*) and its receptors, such as *PLXNA1-4* and *NRP1-2*. Loss-of-function across multiple *SEMA3/NRP/PLXNA* genes can collectively contribute to obesity onset. Case Description: A 3-year-old male was referred for the first time to Outpatient pediatric endocrinology due to early-onset and progressive severe obesity and hyperphagia. He presented neurobehavior disorders and partial diabetes insipidus. At age 6, the child was diagnosed with obesity-related complications, including hyperinsulinemia, impaired glucose tolerance, hypercholesterolemia, hepatic steatosis, and hypovitaminosis. The NGS analysis revealed four variants related to obesity: *SIM1*, *SEMA3C*, *PLXNA4*, and *CREBBP* gene mutations. Conclusions: The case presents the association of SIM-1 gene mutation with other obesity-related variants. The interactive and cumulative effects of the identified variants could coexist in the determination of severe obesity through abnormalities in the development and function of hypothalamic melanocortin circuits related to energy homeostasis. Although the pathogenic mutation of the *SIM1* gene plays the main role, the complex clinical picture may be related to the possible cumulative effect of the other genetic mutations.

## 1. Introduction

Genetics and environmental alterations are responsible for childhood obesity. Oligogenic obesity is caused by the combination of variants in several genes related to obesity and its complications and the influence of environmental factors [1]. Differently, monogenic obesity results from a mutation in a single gene of the leptin–melanocortin pathway and presents with early-onset severe obesity associated with hyperphagia and additional clinical features [2]. Mutations in genes implicated in the leptin–melanocortin pathway are found in 2–5% of patients with monogenic obesity [3]. The presence of mutations in different genes have been reported to be pathogenic for obesity with alterations of the leptin–melanocortin pathway, including leptin (*LEP)*, leptin receptor (*LEPR*), proopiomelanocortin (*POMC*), prohormone convertase 1 (*PCSK1*), and the melanocortin 4 receptor (*MC4R*), [4] as reported in Figure 1. Furthermore, genetic discoveries confirmed the role of other genes in energy homeostasis, such as single-minded homolog 1 (*SIM1*) [5].

The single-minded homolog 1 (*SIM1*) gene encodes for transcription and neurotropic factors that control neuronal differentiation in the hypothalamus. Particularly, SIM1 is implicated in the development of the paraventricular nucleus (PVN) and supraoptic nucleus (SPN) of the hypothalamus, regions that play a critical role in energy homeostasis [6]. The *SIM1* gene is expressed in several areas of the hypothalamus which have a high concentration of *MC4R*-expressing neurons, such as the PVN, basomedial amygdala, anterior hypothalamus, and lateral hypothalamic area [7]. In mice, *SIM1* expression increases following activation of *MC4R* by melanocortin agonists or leptin [8]. *SIM1* haploinsufficiency (*SIM1*+/−) leads to obesity, marked by hyperphagia without reduced energy expenditure, underscoring *SIM1*’s role in energy homeostasis and the leptin–melanocortin–oxytocin pathway [7,8,9]. Mutations in the *SIM1* gene are associated with a wide clinical spectrum. Translocation and missense/frameshift mutations with autosomal dominant inheritance have been related to hyperphagia and early-onset severe obesity [10]. In addition, neurobehavioral alterations including impaired concentration, memory deficits, emotional lability, and autism spectrum behaviors are associated with such mutations [9,10].

The development of melanocortin neural circuits in the hypothalamus is promoted by other factors such as Semaphorine 3 (*SEMA3*) and its receptor, *PLXNA1-4*, and *NRP1-2*. Rare variants were identified across SEMA3A-G and their receptors (*PLXNA1-4*, *NRP1-2*), many of which impaired protein secretion or disrupted signaling via distinct molecular mechanisms [11]. These variants were significantly enriched in severely obese individuals compared to controls, suggesting a strong association with obesity risk. This oligogenic burden, affecting different nodes of the same signaling network, suggests that partial loss-of-function across multiple *SEMA3/NRP/PLXNA* genes can collectively contribute to obesity onset [11].

The authors present the special case of a 12-year-old boy with early-onset severe obesity determined by a pathogenetic mutation in the *SIM1* gene, in which it has been hypothesized how the cumulative effects of other genetic variants detected in melanocortin pathway genes determine the complex clinical picture that includes partial diabetes insipidus, transient metabolic complications, and neurobehavioral disorders.

## 2. Case Presentation

A 3-year-old Caucasian male was referred for the first time to Outpatient pediatric endocrinology due to early-onset and progressive severe obesity and hyperphagia.

He was born full term by a non-consanguineous parent. Gestation was complicated due to placental abruption, and it concluded with cesarean section because of breech presentation. At birth, his Apgar scores were 9 and 10 at 1′ and 5′, respectively. Weight and length at birth were 4100 gr (+1.19 SDS) and 55 cm (+2.43 SDS), respectively; length/weight ratio was appropriate for gestational age.

Family history was significant, for obesity was present both in the patient’s maternal and paternal side. The mother had severe obesity and underwent sleeve gastrectomy due to failure of the dietary-behavioral approach. The father had obesity since childhood; he had a BMI of 32 kg/m^2^ at the time of the visit. Obesity is presented in paternal uncles and grandmother.

Our patient’s clinical history was characterized by an early onset of obesity that progressively worsened and had associations with hyperphagia. In addition, he suffered from neurobehavior disorders and, since the first years of life, he presented significant polyuria and polydipsia.

***Neuropsychiatric assessments.*** During infancy, the patient exhibited neurobehavioral disorders, including motor restlessness, motor stereotypies, irritability with hyperkinetic patterns, and speech disorder, associated with EEG (brain electrical activity in the right parieto-temporal leads) and MRI (reduction in the volume of the left hippocampal formation) abnormalities. The patient underwent speech, psychomotor, and psychological therapies, with recommendations for group recreational activities.

***Nephrological/Endocrinological assessments.*** Since the age of 3 years old, the patient exhibited significant polyuria and polydipsia. A water deprivation test and the administration of desmopressin revealed partial diabetes insipidus (DI). Normal signal intensity and structure of hypophysis were revealed via MRI. Other clinical conditions, including diabetes mellitus, electrolyte imbalances, thyroid dysfunction, urinary tract infections, and iatrogenic causes, were excluded before proceeding with the diagnosis of partial DI. Treatment with desmopressin resulted in a marked reduction in symptoms and complete regression of nycturia.

***Endocrinological/nutritional assessments.*** The patient had early-onset obesity and hyperphagia. At 3 years of age, during his first endocrinological evaluation, his weight was 22 kg (+3.21 SDS), length 99 cm (+0.69 SDS), and BMI 22.3 kg/m^2^ (+2.23 SDS). Physical examination revealed subtle dysmorphic features: narrow palpebral fissures, sunken nasal root, reverse epicanthus, and mild prognathism. Notably, insatiable hunger associated with abnormal food-seeking behaviors was present even during the night hours, with multiple awakenings reported. Despite the typical childhood diet, the progressive worsening of the degree of obesity became more pronounced at the age of 19 months. During endocrinological follow-up, endocrinological dysfunction (e.g., alterations in thyroid function and hypercortisolemia) and syndromic obesity (e.g., Prader–Willi syndrome) were ruled out as causes of obesity.

Despite close dietary-behavioral follow-up, a progressive worsening of the degree of obesity was documented. At 6 years old, the child had a weight of 40.5 kg (+2.90 SDS), length of 122.5 cm (+0.38 SDS), and BMI of 26.9 kg/m^2^ (+2.41 SDS). The waist and hip circumferences were 81 cm (>3 SDS) and 85 cm (>3 SDS), respectively. The child also had obesity-related complications including hyperinsulinemia, impaired glucose tolerance, hypercholesterolemia, hepatic steatosis, and hypovitaminosis D. At age 12 years, the child had a weight of 67.1 kg (+2.23 SDS), BMI of 28.4 kg/m^2^ (+2 SDS), waist circumference of 90 cm, and hip circumference of 97 cm (both > 3 SDS). Growth rate remained regular with a height of 153.8 cm (+0.94 SDS), as well as the progression of pubertal development, which began at age 11 years. As reported in Table 1, there was a regression of the glycol–metabolic alterations with a normalization of glycemic and insulin serum levels that could be justified by a partial change in lifestyle. The nutritional evaluation ended with bioimpedance analysis that revealed a body fat percentage (FAT) of 34.3% and a fat-free mass (FFM) of 65.7% with a basal metabolic rate (BMR) of 1527 Kcal/day.

In view of the family history and clinical phenotype characterized by early-onset obesity associated with behavioral disorders, analysis of 80 obesity-related genes was performed by next-generation sequencing (NGS). The results are reported below.

## 3. Genetic Investigations and Interpretations

The NGS analysis revealed a primary, paternally inherited variant of the *SIM1* gene, c.290dup p.(Asp98Argfs*29), detected in heterozygosity, neither described in the literature nor reported in the ClinVar population database. This is a frameshift variant, within exon 4 (out of 2 exons), that introduces a premature stop codon, which in turn is predicted to lead to the creation of a truncated protein and/or a reduction in its expression by mRNA degradation. According to American College of Medical Genetics and Genomics (ACGM) guidelines, the variant is classified as pathogenic, as reported on the Varsome and Franklin database.

Additionally, other variants are reported and classified, according to the ACGM, as of uncertain significance (VOUS), as reported in Figure 2:-paternally inherited variant in the *SEMA3C* gene, c.915A>C p.(Leu305Phe).-paternally inherited variant in the *PLXNA4* gene, c.3041T>A p.(Val1014Glu).-maternally inherited variant in the *CREBBP* gene, c.437C>T p.(Ala146Val).

The *SAMA3C* gene variant, detected in heterozygosity, is a missense variant not described in the literature but reported in the population database gnomAD. According to ACGM guidelines, the variant is considered as VUOS on the Franklin database and as likely pathogenic on the Varsome database.

The *PLXNA4* gene variant, detected in heterozygosity, is not described in the literature, and is not reported in the population database ClinVar. According to ACGM guidelines, this missense variant is classified as likely pathogenic, as reported on the Varsome and Franklin databases.

*PLXNA4* encodes the protein Plexin-A4, a member of the plexin family. Plexins function as receptors for semaphorins, a large family of proteins, including *SAMA3C*. Plexin-A4 has been specifically implicated in guiding nerve growth and neural development. Mutations in the *PLXNA4* gene may impair the proper response to hunger and satiety signals, thereby exacerbating the effects of other mutations [11].

The *CREBBP* gene, detected in heterozygosity, is the only variant maternally inherited. This is a missense variant that is not described in the literature but reported in the population database ClinVar, and, according to ACGM guidelines, this missense variant is classified as VOUS, as reported on the Varsome and Franklin databases.

Next generation sequencing (Illumina, Centro de Genética Clinica e Patologia S.A, Rua Sa da Bandeira 706, Portugal) was performed on genomic DNA upon the capture of target regions of whole exome using oligonucleotide probes. After alignment, base/CNV calling and annotation was performed using the reference genomes *Homo sapiens* (GRCh38), SNVs (single nucleotide variants), and indels, and CNVs (copy number variations) were filtered, and a structured analysis was performed to assess their pathogenicity and potential to explain the clinical phenotype. In a single gene analysis, if necessary, Sanger sequencing was used to provide data for bases with insufficient coverage (minimum coverage of less than 15×). The variants detected in the flanking exons and intronic regions (±10 bp) of the genes mentioned in the list referred below were evaluated.

The NGS panel of obesity-related genes used in the analysis included the following genes: ADCY3, AFF4, ALMS1, ARL6, BBIP1, BBS1, BBS10, BBS12, BBS2, BBS4, BBS5, BBS7, BBS9, BDNF, CEP290, CFAP418, CPE, CREBBP, CUL4B, DNMT3A, DYRK1B, EP300, GNAS, HTR2C, IFT172, IFT27, IFT74, INPP5E, ISL1, KIDINS220, KSR2, LEP, LEPR, LZTFL1, MAGEL2, MC3R, MC4R, MECP2, MKKS, MKS1, MRAP2, NCOA1, NR0B2, NRP1, NRP2, NTRK2, PCNT, PCSK1, PHF6, PHIP, PLXNA1, PLXNA2, PLXNA3, PLXNA4, POMC, PPARG, PROK2, RAB23, RAI1, RPGRIP1L, RPS6KA3, SDCCAG8, SEMA3A, SEMA3B, SEMA3C, SEMA3D, SEMA3E, SEMA3F, SEMA3G, SH2B1, SIM1, TBX3, TRIM32, TRPC5, TTC8, TUB, UCP3, VPS13B, and WDPCP + 16p11.2

In addition, comparative genomic hybridization (CGH)-array analysis identified a de novo microduplication of the 16p11.2 region (OMIM#614671), approximately 564 kb in size, which is associated with an increased predisposition to behavioral disorders and autism spectrum disorders.

Summary of the genetic variants identified is reported in Figure 2 and Table 2.

## 4. Discussion

This case report describes the clinical history of a child with early onset of severe worsening obesity related to a pathogenic mutation of the *SIM1* gene. The peculiarity of this case is related not only to the rarity of the *SIM1* gene pathogenic mutations, but also to the possible cumulative effect of the other mutations the patient carries in the expression of the clinical picture. The interactive and cumulative effects of the identified variants could coexist in the determination of severe obesity through abnormalities in the development and function of hypothalamic melanocortin circuits related to energy homeostasis. Although functional and genetic studies investigating the combined effects of mutations in both the *SEMA3-PLXNA* pathway and *SIM1* have not yet been conducted, the convergence of their roles in hypothalamic development strongly suggests a synergistic impact on obesity risk. Disruptions in *SEMA3* signaling may hinder the formation of melanocortin circuits, while *SIM1* mutations may compromise the structural integrity and function of the paraventricular nucleus (PVN). Critically, the simultaneous impairment of these pathways could interact in a compounding manner, severely disrupting hypothalamic regulation of energy balance and dramatically increasing susceptibility to severe obesity in this patient.

According to the Endocrine Society clinical practice guidelines [12], genetic analysis is recommended in patients with early-onset obesity, before 5 years of age, and with clinical features suspected of genetic disorders, such as extreme hyperphagia and/or a family history of severe obesity. Our patient presents clinical features suspected of genetic obesity, including a familiar history of severe obesity, an early onset of obesity, hyperphagia, and neurobehavioral abnormalities. The hypothalamus plays a crucial role in regulating satiety and resting energy expenditure [13]. The progressive worsening of the degree of obesity and the persistence of hyperphagia must be considered red flags, as they could imply an alteration of hypothalamic pathways in the regulation of energy balance. Genetic causes of hypothalamic dysfunction (HD) are due to mutations in genes implicated in the leptin–melanocortin pathway [14]. Autosomal dominant *SIM1* gene mutations cause monogenic non-syndromic hypothalamic obesity with a primary mechanism characterized by the disruption of the hypothalamic development, particularly of the PVN and SON, with a reduction in *MC4R* expression and OXT, AVP, corticotropin-releasing hormone (CRH), thyrotropin-releasing hormone (TRH), and somatostatin (SS) neurons [15]. In vitro, *SIM1* factor is a crucial molecular mechanism in a specific melanocortin-signaling pathway that regulates food intake independently of energy expenditure and without interfering with the thermogenesis pathway [16]. The *SIM1* gene is predominantly involved in satiety alertness (thus hyperphagia) and resting energy expenditure, although the details regarding *SIM1* gene activity are still not completely clear [9]. In addition, despite the molecular targets of *SIM1* not being completely known, the anorexigenic neuropeptide oxytocin mediates part of the actions of *SIM1*, which lies downstream of *MC4R*. Oxytocin neurons in the PVN are stimulated by an *MC4R* agonist in mice, and the hyperphagia in *SIM1*-haploinsufficient mice is reduced after the central administration of oxytocin (Figure 1) [8]. *SIM1* acts downstream of *MC4R*, and both can cause monogenic obesity via autosomal dominant inheritance. The *MC4R* gene mutation is the most frequent cause of monogenic obesity, and it shares several clinical aspects with forms of multifactorial obesity [17]. However, patients with *SIM1* gene mutation present different clinical features from patients with *MC4R* variants. While neurobehavioral disorders are present in patients with the *SIM1* gene mutation, they are not reported in patients with the *MC4R* gene mutation [9,18]. Patients with the *MC4R* variant have better accelerated linear growth in the pre-pubertal phase and greater final height than patients with *SIM1* gene mutations [18]. Our patient presented a regular growth rate, and he developed neurobehavior disorders. These clinical differences suggest that *MC4R–SIM1* interaction does not occur in an exclusively linear pathway, despite *SIM1* lying downstream of *MC4R*. Future molecular characterization is necessary to know the correct action pathway of *SIM1*.

Chromosomal deletions involving the 6q14–q21 region, which encompasses several genes including *SIM1,* have been associated with early-onset obesity and developmental delay [19,20]. Some studies have also suggested a phenotypic overlap with Prader–Willi syndrome (PWS) among patients with point mutation of *SIM1* [21,22,23]. As in our patient, some of these features overlap with those reported in PWS, but crucial hallmarks of PWS were absent, such as a history of hypotonia or feeding difficulties in early life [9,23].

Glucometabolic complications and metabolic syndrome are not reported systematically in patients with *SIM*-gene alteration in childhood and adulthood [23,24]. Ramachandrappa et al. reported hyperinsulinemia in all the patients evaluated and impaired glucose tolerance in one case [9], while Stanikova et al. reported the presence of metabolic syndrome in the proband but not in the other family members carrying the same mutation (p.D134N variant of the *SIM1* gene) [25,26].

An autosomal dominant variant in the *SIM1* gene could have implications in the development of vasopressin (AVP), corticotropin-releasing hormone (CRH), and thyrotropin-releasing hormone (TRH) neurons. These potentials roles are supposed in several mice studies [27], while hypopituitarism seemed to be due to the association of *SIM1* with other variants, such as *POU3F2* gene mutations [6]. Hypopituitarism was reveled in a 21-month-old male with early-onset obesity associated with central hypothyroidism, central adrenal insufficiency, and partial diabetes insipidus. The patient presented a novel *SIM1* gene mutation (c.214C>T (p.Pro72Ser)). Our patient, instead, suffered from partial diabetes insipidus because of a progressive polydipsia since his first years of life, which is considered as a potential condition of hypothalamic disfunction. Furthermore, despite little evidence, an evaluation of pituitary function should be considered in patients with severe obesity and genetic abnormality, especially regarding the *SIM1* gene [28].

Mutation of *SIM1* is not always responsible for a fully penetrant form of obesity. *SIM1* variants segregate with obesity in extended family studies with variable penetrance [23]. Several missense variants of *SIM1* may underpin a monogenic form of obesity. Instead, other missense variants cause a variable loss-of-function of the *SIM1* and co-segregate with being overweight/obesity. In addition, both environmental factors and genetic background play a role in the penetrance of *SIM1*-related pathology [23,29].

*SEMA3C* encodes a protein belonging to the semaphorin family. *PLXNA4* encodes the protein Plexin-A4, a member of the plexin family. Plexins are receptors of semaphorins. Semaphorins and their receptors play a critical role in axonal guidance, neuronal migration, the development of hypothalamic neuronal circuits, angiogenesis, and immune responses [11]. Mutations in the *SEMA3C* gene may disrupt the connectivity between the (PVN), potentially exacerbating dysregulation of appetite. Plexin-A4 has been specifically implicated in guiding nerve growth and neural development. Mutations in the *PLXNA4* gene may impair the proper response to hunger and satiety signals, thereby exacerbating the effects of other mutations. Semaphorin signaling encourages the development of hypothalamic melanocortin circuits. Forty mutations in *SEMA3A-G* and their receptors (*PLXNA1-4*; *NRP1-2*) were detected in 573 severely obese patients [11]. The interaction between *SEMA3* and *PLXNA* ensures the innervation of the PVN by arcuate POMC axons. *SEMA3C* participates in the growth of arcuate NPY axons and loss of function of *SEMA3*s receptors (*PLXNA1-4*; *NRP1-2*) in POMC neurons, and disrupts arcuate POMC projections to the PVN, and thus has caused an observed reduction in energy expenditure and weight gain in mice and zebrafish [11,30]. Also, *CREBBP* is related to obesity, and it is a transcriptional coactivator involved in neuroplasticity and the regulation of metabolism in the hypothalamus.

While the heterozygosis *SIM1* gene mutation is classified as pathogenic according to ACGM, the other heterozygosis variants of the patients are considered as variants of uncertain significance, VOUS [31]. Despite the presence of an elevated number of VOUS, the molecular diagnosis of monogenic obesity is becoming more accurate over time. Genetic re-interpretation of the VOUS variants should be characterized by a reevaluation of the significance of the variants over time. Future evidence could be decisive in the reinterpretation of the significance of the three VOUS variants detected in our patient [31].

These findings highlight the critical importance of comprehensive genetic assessment that includes not only clearly pathogenic variants, such as the one identified in SIM1, but also variants of uncertain significance in other genes functioning within the same closely related biological pathway. Such VOUSs, though individually insufficient to cause disease, may act as genetic modifiers that potentiate the deleterious effects of a primary pathogenic mutation, thereby exacerbating the clinical phenotype and contributing to variability in disease severity.

Prospectively, given the pathogenetic mechanism underlying obesity-related genes *SIM1* and *SEMA3-PLXNA*, the therapeutic role of setmelanotide can be hypothesized, and is currently being evaluated (ClinicalTrials.gov ID: NCT04963231, NCT05093634). Nevertheless, the first therapeutic approach should be the dietary-behavioral approach, which in our patient was able to slow down the worsening degree of obesity and, moreover, resulted in regression of metabolic complications.

The interpretation of the aspects described in this manuscript should be considered in the light of the limitation of only a single case being reported. Furthermore, no data are yet available on functional analyses or gene expression information regarding the combined effects of mutations in both the SEMA3–PLXNA and SIM1 pathways. Future genetic studies are needed to further clarify the combined effect of the described variants.

## 5. Conclusions

In conclusion, to the best of our knowledge, this case presents for the first time the association of *SIM1* gene mutation with other obesity-related variants in a patient with early obesity, hyperphagia, behavioral disorders, and partial diabetes insipidus. The pathogenetic mutation of the *SIM1* gene evidently plays the main role in our patient’s obesity condition. However, possible interactions with the described variants could jointly promote alterations in hypothalamic function.

## Figures and Tables

**Figure 1 genes-16-00588-f001:**
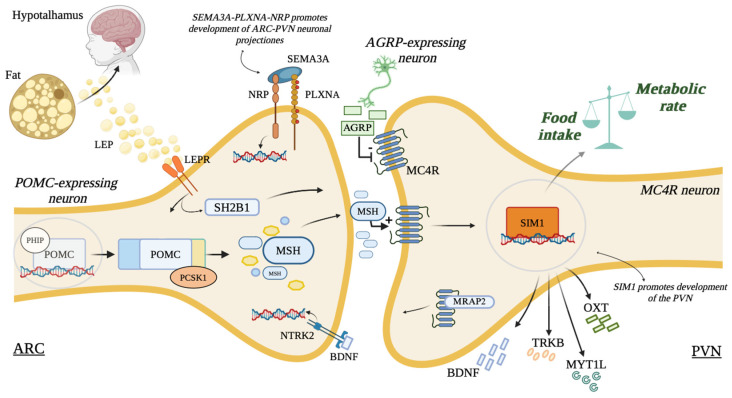
Leptin–melanocortin pathway and molecular role of SIM-1. The leptin–melanocortin pathway involves neurons in the arcuate nucleus (*ARC*) of the hypothalamus that either express pro-opiomelanocortin (*POMC*) or agouti-related protein (*AGRP*). These neurons detect circulating leptin levels, which correspond to body fat. In response, they communicate with neurons in the paraventricular nucleus (PVN) that express the melanocortin 4 receptor (*MC4R*), helping to regulate appetite and connect energy reserves to eating behavior. The growth and direction of *POMC* neuron connections to the PVN are influenced by class 3 semaphorins (*SEMA3*) binding to neuropilin co-receptors (*NRP*) and Plexin-A (*PLXNA*) receptors. Leptin may also promote synaptic changes through brain-derived neurotrophic factor (*BDNF*) binding to its receptor neurotrophic receptor tyrosine kinase 2 (*NTRK2*), affecting ARC and PVN neurons. Proper development of the PVN depends on the transcription factor SIM1. Myelin transcription factor 1 like (*MYT1L*), BDNF, tropomyosin receptor kinase B (*TRKB*), and oxytocin (*OXT*) are potential targets of SIM1 in the energy homeostasis. MRAP2: melanocortin 2 receptor accessory protein 2; MSH: melanocyte stimulating hormone; PCSK1: prohormone convertase 1; PHIP: pleckstrin homology domain interacting protein; SH2B1: SH2B adaptor protein 1. Created in Biorender by Giovanni Luppino. https://app.biorender.com/user/signin.

**Figure 2 genes-16-00588-f002:**
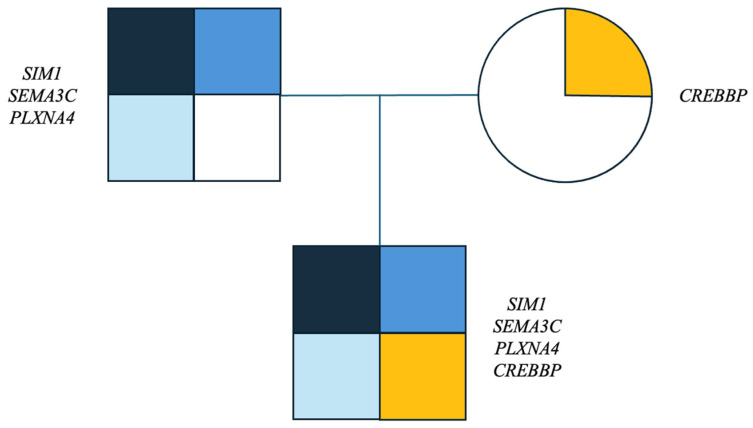
Pedigree of the family and the variants in each individual. Each colour represents a single mutation: dark blue represents the SIM1 gene mutation, medium blue the SEMA3C mutation, light blue the PLXNA4 gene mutation, yellow the CREBBP gene mutation.

**Table 1 genes-16-00588-t001:** Clinical and main biomolecular aspects of the patient at 6 years old and 12 years old, respectively.

	6 Years Old	12 Years Old
Weight (kg)	40.5	67.1
Weight (SDS)	2.90	2.23
Length (cm)	122.5	153.8
Length (SDS)	0.38	0.94
Body mass index (kg/m^2^)	26.9	28.37
Body mass index (SDS)	2.41	2
Tanner stage	G1P1	G3P3
Glucose (mg/dL)	111	91
Glucose (mg/dL) at 30′ min	129	134
Glucose (mg/dL) at 60′ min	198	127
Glucose (mg/dL) at 90′ min	194	139
Glucose (mg/dL) at 120′ min	155	122
Insulin (uIU/mL)	26.3	14.7
Insulin (uIU/mL) at 30′ min	63.1	70.9
Insulin (uIU/mL) at 60′ min	51.6	72.3
Insulin (uIU/mL) at 90′ min	118	92.8
Insulin (uIU/mL) at 120′ min	33.3	80.2
HbA1c (%)	4.5%	4.3%
Total Cholesterol (mg/dL)	213	174
LDL (mg/dL)	100	90
HDL (mg/dL)	83	65
Triglycerides (mg/dL)	44	54
ACTH (pg/mL)	26.7	21.9
Cortisol (ug/dL) at 08:00 a.m.	17.3	7.61
TSH (uIU)/mL	1.53	1.520
FT4 (pmol/L)	17.5	14
Steatosis	Yes	No

**Table 2 genes-16-00588-t002:** Genetic variants identified in proband.

Gene	MOI	Variant Type	Variant	gnomAD/DGV	Classification (ACGM)	Zygosity
*SIM1*	AD	Frameshift	NM_005068.3:c.290dup p.(Asp98Argfs*29)	-	Pathogenic	Het
*SEMA3C*	AD	Missense	NM_006379.5:c.915A>C p.(Leu305Phe)	0.0032% 1 het	VOUS (suspected pathogenic)	Het
*PLXNA4*	AD	Missense	NM_020911.2:c.3041T>A p.(Val1014Glu)	-	VOUS (suspected benign)	Het
*CREBBP*	AD	Missense	NM_004380.3:c.437C>T p.(Ala146Val)	-	VOUS (suspected benign)	Het

MOI—Mode of Inheritance; AD—Autosomal dominant; Het—Heterozygosity. Minor allele frequency for all populations combined Genome Aggregation Database (gnomAD) version v2.1.1 and Database of Genomic Variants (DGV).

## Data Availability

The original contributions presented in this study are included in the article. Further inquiries can be directed to the corresponding author.

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
