# Peer review of "Cumulative Effects of Genetic Variants Detected in a Child with Early-Onset Non-Syndromic Obesity Due to SIM-1 Gene Mutation"

_genes, 2025, doi:10.3390/genes16050588_

Round 1
Reviewer 1 Report
Comments and Suggestions for Authors
- In section 2 case presentation, please describe the race/ethnicity of the patient.
- Under the section genetic investigations and interpretations. Please describe the NGS analysis in details. What assay, and what bioinformatics pipeline was used to detect the genetic mutations.
- in line 167, should add the full name of the ACGM abbreviation. In line 170, ACGM full name should not be included.
- It might be useful to include the functional analyses or gene expression information of the four mutations in genes SIM1, SEMA3C, PLXNA4, and CREBBP.
- please describe some potential limitations of your study in the discussion section.
< !-- notionvc: 5f2fbf8a-8ead-47f7-b7d6-6460d19a86e4 -->
Reviewer 2 Report
Comments and Suggestions for Authors
I have no significant comments on this manuscript. This is a typical case report. All the elements required for this type of publication have been maintained. The case is interesting. The research was conducted according to typical analytical-medical procedures. The discussion is well written. References to relevant literature sources have been made. Conclusions are logical and related to the described case. The format of the references does not comply with the MDPI guidelines and requires correction.
Author Response
Answers to the comments of the Reviewer.
We are very grateful to this Reviewer for the revision and opinion of our manuscript.
Comments:
I have no significant comments on this manuscript. This is a typical case report. All the elements required for this type of publication have been maintained. The case is interesting. The research was conducted according to typical analytical-medical procedures. The discussion is well written. References to relevant literature sources have been made. Conclusions are logical and related to the described case. The format of the references does not comply with the MDPI guidelines and requires correction.
Thanks to the reviewer for your opinion. We have corrected the format of the references according to MDPI guidelines.